# Exploring menstrual products: A systematic review and meta-analysis of reusable menstrual pads for public health internationally

**Anna Maria van Eijk**[1]*, **Naduni Jayasinghe**[1], **Garazi Zulaika**[1], **Linda Mason**[1], **Muthusamy Sivakami**[2], **Holger W. Unger**[1,3,4], **Penelope A. Phillips-Howard**[1]

1 Department of Clinical Sciences, Liverpool School of Tropical Medicine, Liverpool, United Kingdom, 2 Tata Institute of Social Sciences, Mumbai, India, 3 Department of Obstetrics and Gynaecology, Royal Darwin Hospital, Darwin, Australia, 4 Menzies School of Health Research, Charles Darwin University, Darwin, Australia

* Anna.vanEijk@lstmed.ac.uk

**Data Availability Statement:** All relevant data are within the manuscript and its Supporting Information files.

## Abstract

### Background

Girls and women need effective, safe, and affordable menstrual products. Single-use menstrual pads and tampons are regularly provided by agencies among resource-poor populations. Reusable menstrual pads (RMPs: fabric layers sewn together by an enterprise for manufacture of menstrual products) may be an effective alternative.

### Methods

For this review (PROSPERO CRD42020179545) we searched databases (inception to November 1, 2020) for quantitative and qualitative studies that reported on leakage, acceptability, or safety of RMPs. Findings were summarised or combined using forest plots (random-effects meta-analysis). Potential costs and environmental savings associated with RMPs were estimated.

### Results

A total of 44 studies were eligible (~14,800 participants). Most were conducted in low- and middle-income countries (LMIC, 78%), and 20% in refugee settings. The overall quality of studies was low. RMP uptake in cohort studies ranged from 22–100% (12 studies). One Ugandan trial among schoolgirls found leakage with RMPs was lower (44.4%, n = 72) compared to cloths (78%, n = 111, p<0.001). Self-reported skin-irritation was 23.8% after 3 months among RMP-users in a Ugandan cohort in a refugee setting (n = 267), compared to 72.8% at baseline with disposable pad use. There were no objective reports on infection. Challenges with washing and changing RMP were reported in LMIC studies, due to lack of water, privacy, soap, buckets, and sanitation/drying facilities. Among 69 brands, the average price for an RMP was $8.95 (standard deviation [sd] $5.08; LMIC $2.06, n = 10, high-

**Funding:** This study is funded by the Joint Global Health Trials Initiative (UK-Medical Research Council/Department for International Development/ Wellcome Trust/Department of Health and Social Care, grant MR/N006046/1). The funders had no role in the design of the study, the collection, analysis, and interpretation of data, or in writing the manuscript. The corresponding author had full access to all data in the study and had final responsibility to submit for publication.

**Competing interests:** The authors have declared that no competing interests exist.

income countries [HIC] $10.11), with a mean estimated lifetime of 4.3 years (sd 2.3; LMIC 2.9, n = 11; HIC 4.9 years, n = 23). In 5-year cost-estimates, in LMICs, 4–25 RMPs per period would be cheaper (170–417 US$) than 9–25 single-use pads, with waste-savings of ~600–1600 single-use pads. In HICs, 4–25 RMPs would be cheaper (33–245 US$) compared to 20 single-use tampons per period, with waste-savings of ~1300 tampons.

## Conclusion

RMPs are used internationally and are an effective, safe, cheaper, and environmentally friendly option for menstrual product provision by programmes. Good quality studies in this field are needed.

## Introduction

Girls, women, and transgender people have struggled throughout history to combine menstruation with daily life; however, this struggle is generally invisible [1]. Most girls start menstruating between 12 and 14 years [2], which is a pivotal time centred on biopsychosocial development and education [2–4]. On average a woman will spend 65 days per year menstruating [5]. In a survey among European countries, 60% of interviewed women would prefer menstruation to be less frequent than once a month, with quality-of-life considerations given as the main reasons [6]. Few options are available to manage menstruation; in high-income countries (HICs), tampons and single-use pads are commonly used. Menstrual cups, commercially available reusable pads (RMP: layers of fabric sewn together as a period pad in an enterprise for production of menstrual products), and period pants are less-known alternatives [7]. Tampons are less frequently used in low- and middle-income countries (LMIC) [8–10]; however, use of single-use pads is common as is the use of non-commercial cloths that can be reused or disposed, and a whole range of other non-hygienic makeshift materials in times of dire need [11,12]. Adequate options to deal with menstruation allow girls and women to continue their activities, work, or education without fear of leakage [13,14]. Ideally, menstrual products should be comfortable and not result in a reduction of mobility, injuries to the perineum, vulva and vagina, or genitourinary tract and skin infections. Considerations for choice of product include cost, access, ease of use, method of disposal, water and sanitation facilities for changing and washing, and resulting environmental impacts caused by the selected product. Ignorance, prejudice, cultural norms, lack of means, setting, safety fears, and lack of availability can impede girls and women from testing the full range of products available to assess what works best for them to manage their menstruation.

In several countries, the number of policy-led initiatives and donations to provide menstrual products, or tax bans on menstrual products have increased recently, e.g., to allow girls to attend school, to assist impoverished women, or to achieve gender-equity [8,15–18]. Studies including trials in low-resource settings also provide cash for girls to purchase menstrual products, further increasing the need for a review of the effectiveness, use, and safety of products available for menstruation [19,20]. A review of commercially available products will inform women, girls, and programme and policy-decision makers on product choices. To document current knowledge on available reusable products, we recently reviewed the menstrual cup [7]. In this systematic review and meta-analysis, we review what is known about the effectiveness, safety, acceptability, availability, costs, and waste of RMPs.

## Methods

### Search strategy and selection criteria

We searched PubMed, Cochrane Library, Web of Science, Medline, Global Health database, Cinahl, Science.gov and WorldWideScience, and Google Scholar for material from the inception of the database until 1 November 2020 using the keywords (cloth* OR towel* OR pad OR suppl* OR product* OR absorbent*) AND (menses OR menstrual OR menstruation) AND (recyclable OR reusable OR sustainable). Additional information on the search can be found in the supplement (S1 File). We searched the reference lists of relevant studies, websites of professional bodies, non-governmental organisations and grey literature (e.g., reports or conference abstracts) and contacted experts in the field to recommend relevant reports. Study eligibility, data extraction, and risk-of-bias assessment were done independently by two reviewers (AMvE and NJ for quantitative and LM and GZ for qualitative studies); a third person acted as tiebreaker if discussions could not resolve differences (PPH). Cloths, defined as home-made pieces of material used to absorb menstrual blood which can be disposed, or cleaned and reused were differentiated from commercial reusable menstrual products, which are layers of fabric sewn together by an enterprise for production of menstrual products (e.g., commercial reusable pads, period underwear, labia pads; this will be summarized as reusable menstrual pads or RMPs). In this review, we focused on commercially available and not-for-profit products produced by non-governmental organizations and excluded home-made reusable pads or cloths. To be eligible for inclusion, the reports needed to have information on use, safety, effectiveness, efficacy, or acceptability of RMPs. The main outcome of interest was menstrual blood leakage. Additional outcomes of interest were acceptability and ease of use, including washing and drying, and comfort of wearing. Safety outcomes included rashes, itching, burning, chaffing, or genitourinary infections and any other adverse event reported. We screened websites with education material on menarche for the mention of alternative menstrual items such as RMPs and screened websites of sellers of RMPs to assess costs and materials used (see further S1 File).

### Data analysis

For quality and bias assessments, we used the Cochrane tool for trials, an adaptation of the Newcastle-Ottawa tool (S1 File) for observational studies and the Critical Appraisal Skills Programme (CASP) tool for qualitative studies. We tabulated our findings as a narrative synthesis, and calculated p-values for comparisons where participants belonged to distinct groups using the chi-square test for categorical outcomes and the t-test for continuous outcomes. For cohort studies with a baseline and endline evaluation, a chi-square test is not appropriate because of the repeated measurements in (part of) the population; a McNemar would be appropriate but generally studies provided insufficient information to be able to conduct statistical testing. If studies presented sufficiently homogeneous data in terms of design and outcome, we pooled results using meta-analysis and a random-effects model with heterogeneity quantified using the $I^2$ statistic (S1 File). We examined the following sources of heterogeneity if sufficient data was available using subgroup analysis: setting of the study (high-income *vs* low-income and middle-income countries), study population (adult women *vs* adolescents), year of study (study conducted before or after 2000), and type and duration of RMP used. When we assessed the generally non-random enrolment of participants as too heterogeneous, we showed the results in a forest plot but did not summarize the overall results as a pooled estimate. Qualitative data were analysed using thematic synthesis as described by Thomas and Harden (2008) [21], through which key themes were identified (for further details see S1 File). We integrated

the quantitative and qualitative analyses for the acceptability of RMPs. We used estimates on mean costs from previous calculations for single-use pads, tampons and menstrual cups [7]. We compared estimates of costs of menstrual products using different estimates of numbers of items needed, and the lifespan of the RMP. Additional information on methods used, availability and prices, qualitative studies, and costs and waste, and additional information on data extraction are in the supplement (S1 File). We used two-tailed p values of less than 0.05 to indicate statistical significance. We used Metaprop (Stata version 14.2.2) for the statistical analyses. This systematic review is registered on PROSPERO (CRD42020179545).

## Results

The searches resulted in 353 items of interest after removing duplicates; 212 were excluded after screening and the full text was obtained for 141 items (Fig 1). Fifty-two items (31 articles, 9 reports, 8 theses, and 4 other materials) covering 44 studies, were retained. These studies

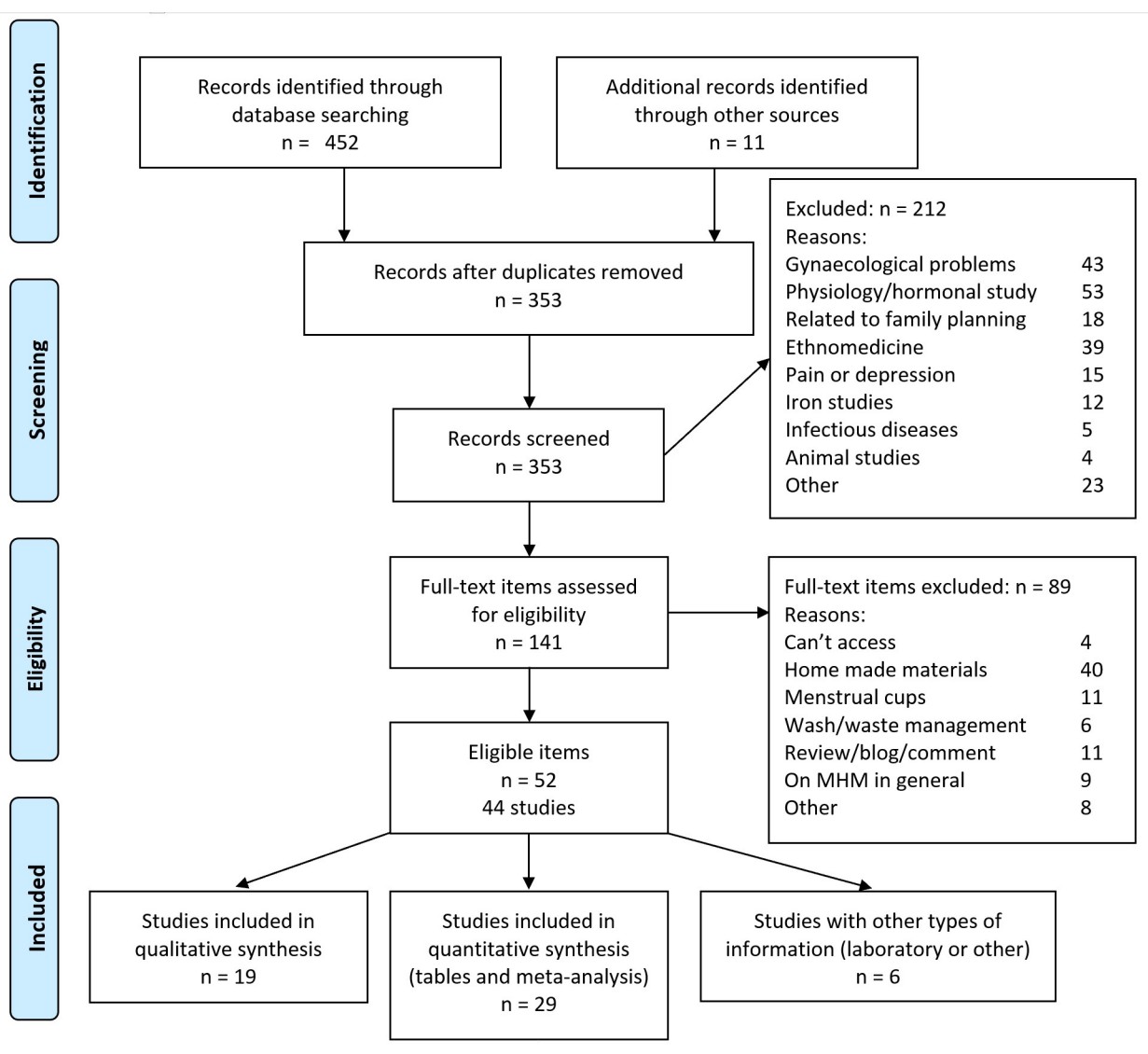

**Fig 1. PRISMA flow diagram.**

**Table 1A. Characteristics of trials contributing to the Reusable Menstrual Pad review (alphabetical order).**

| | Studies | Material | Location | Time | Design | Sample size and population | Information age in years | RMP brand (number received) | Comparison | Follow up | Outcomes | Quality score |
|---|---|---|---|---|---|---|---|---|---|---|---|---|
| 1 | Montgomery 2016 [23] Hennegan 2016 [24] Hennegan 2016 [25] Hennegan 2017 [26] | Journal article | Uganda, Kamuli district | 2012–2014 | Cluster quasi-randomized trial | 1124 schoolgirls (8 primary schools in 4 clusters) | Mean 11.4 yrs, sd 1.7, n = 281 | AFRIpad (6)* | Usual item, not specified Arms: Afripad & education, no education, education only, and none | 24 months | School attendance Psychosocial well-being Use and acceptability | 1 |
| 2 | Garikipati 2019 [27] | Report | India, Hyderabadslums | 2017–2018 | Stratified random allocation | 293 women living in slums | Mean 28.0, sd 7.5, n = 277. Range: 18–45 | Safepad (4) | Compostable disposable pad (Anandi pad), and education on sustainable menstrual material only | 6 months | Women's preference for sustainable menstrual material | 2 |

For details of brands mentioned in this table, see S1 File.

were conducted in 20 countries (four HIC); 31% of studies were in Uganda and 20% were in refugee settings (Table 1, S1 File for qualitative studies). Not all studies reported exact sample sizes, but they involved at least 14,812 participants, and the majority were schoolgirls (9736 or 66%). All quantitative studies were assessed as of low-to-moderate quality (S1 File). In cohort studies loss-to-follow up was either high or not reported; surveys did not report refusal rates and were generally convenience samples (S1 File). Six (31.6%) of the qualitative studies were assessed to be high quality (S1 File). Only one qualitative study involved period underwear, all others involved reusable pads [22]. Details of the RMPs used in these studies, when known, are available in the supplement (S1 File).

## Use and uptake of RMPs

Twenty-one studies provided information on use of RMPs in surveys or at enrolment in cohort studies (Fig 2). Overall, use was low, ranging from 0–88% in LMICs (median 12.5%) and 0–19% in HICs (median 9.4%). The pooled estimate among schoolgirls in Uganda, a more homogenous subgroup, was 13% (95% CI 7–21%, 5 studies, $I^2$ 96.4%, S1 File). Higher use was present in areas where there was a history of a programme that had offered reusable pads, such as in refugee camps [42,53] or in schools [8]. Fifteen longitudinal studies followed participants after distributing RMPs, with a median follow-up time of six months (range 2–18 months) (Table 2). Information on number of participants at follow up and uptake of RMPs was available for 12 studies in 17 locations, all in LMICs; median uptake at follow-up was 90% (range 22–100, Fig 3; the pooled uptake in 6 studies in schools in Uganda was 72%, 95% CI 51–89%, $I^2$ 96.4%, S1 File). There was no correlation between uptake and length of follow up time (Pearson's r = 0.0277, p = 0.9103; S1 File). In three cohort studies involving RMPs, a second/alternate reusable product (menstrual cups) was given to a different group of participants [30,31], or together with the RMPs [31,41]; uptake of RMPs was consistently higher than for menstrual cups (e.g. after six months, use of RMPs was 96% vs. menstrual cups 65% in Tanzania [30]; after four months, use of RMPs was 100% vs. 61% menstrual cups in Uganda [31]). To understand factors associated with uptake of RMPs, a study in Uganda is notable: uptake of RMPs in a local primary school was 100% among girls who reported they used cotton wool for

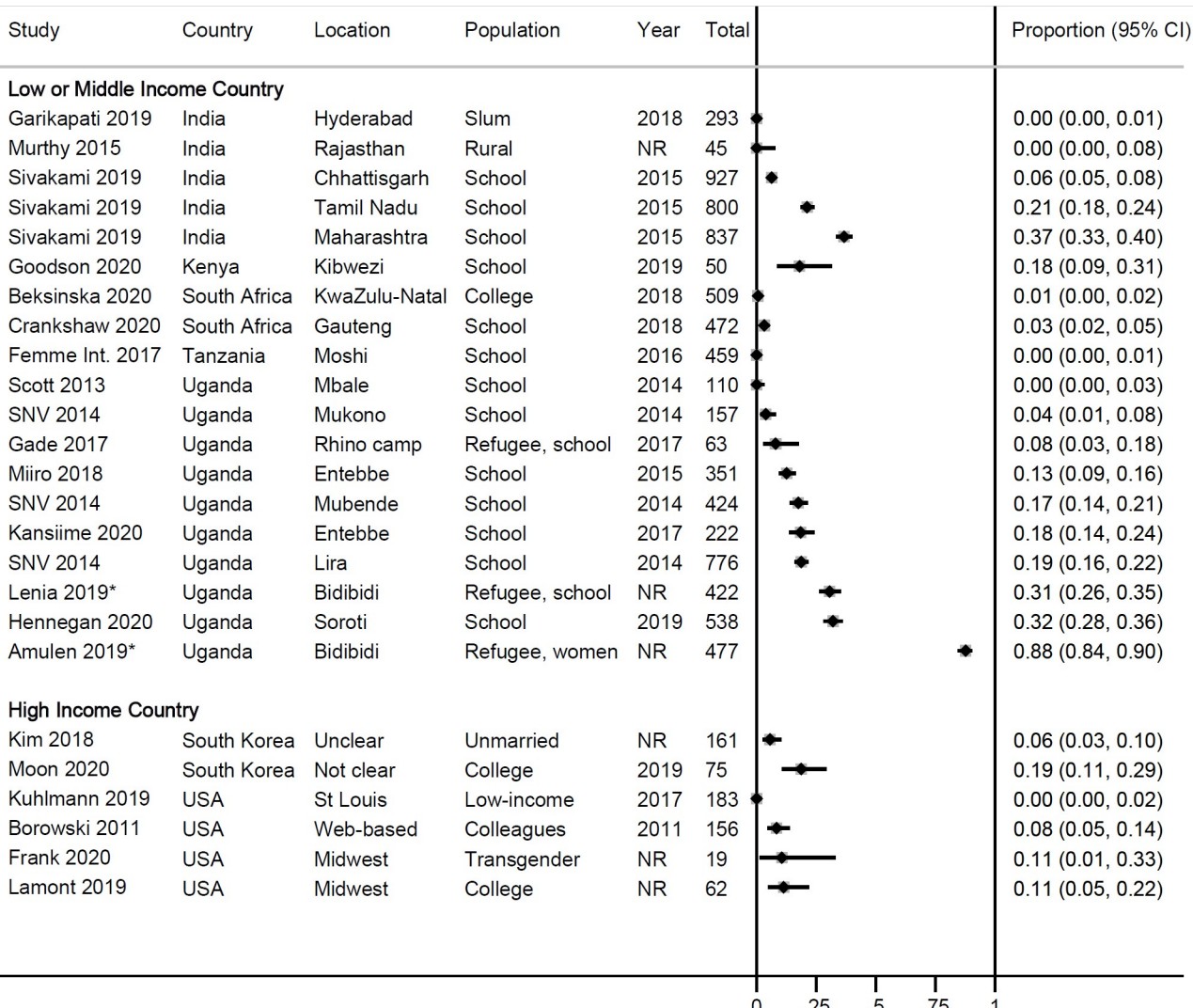

| Study | Country | Location | Population | Year | Total | Proportion (95% CI) |
|---|---|---|---|---|---|---|
| **Low or Middle Income Country** | | | | | | |
| Garikapati 2019 | India | Hyderabad | Slum | 2018 | 293 | 0.00 (0.00, 0.01) |
| Murthy 2015 | India | Rajasthan | Rural | NR | 45 | 0.00 (0.00, 0.08) |
| Sivakami 2019 | India | Chhattisgarh | School | 2015 | 927 | 0.06 (0.05, 0.08) |
| Sivakami 2019 | India | Tamil Nadu | School | 2015 | 800 | 0.21 (0.18, 0.24) |
| Sivakami 2019 | India | Maharashtra | School | 2015 | 837 | 0.37 (0.33, 0.40) |
| Goodson 2020 | Kenya | Kibwezi | School | 2019 | 50 | 0.18 (0.09, 0.31) |
| Beksinska 2020 | South Africa | KwaZulu-Natal | College | 2018 | 509 | 0.01 (0.00, 0.02) |
| Crankshaw 2020 | South Africa | Gauteng | School | 2018 | 472 | 0.03 (0.02, 0.05) |
| Femme Int. 2017 | Tanzania | Moshi | School | 2016 | 459 | 0.00 (0.00, 0.01) |
| Scott 2013 | Uganda | Mbale | School | 2014 | 110 | 0.00 (0.00, 0.03) |
| SNV 2014 | Uganda | Mukono | School | 2014 | 157 | 0.04 (0.01, 0.08) |
| Gade 2017 | Uganda | Rhino camp | Refugee, school | 2017 | 63 | 0.08 (0.03, 0.18) |
| Miiro 2018 | Uganda | Entebbe | School | 2015 | 351 | 0.13 (0.09, 0.16) |
| SNV 2014 | Uganda | Mubende | School | 2014 | 424 | 0.17 (0.14, 0.21) |
| Kansiime 2020 | Uganda | Entebbe | School | 2017 | 222 | 0.18 (0.14, 0.24) |
| SNV 2014 | Uganda | Lira | School | 2014 | 776 | 0.19 (0.16, 0.22) |
| Lenia 2019* | Uganda | Bidibidi | Refugee, school | NR | 422 | 0.31 (0.26, 0.35) |
| Hennegan 2020 | Uganda | Soroti | School | 2019 | 538 | 0.32 (0.28, 0.36) |
| Amulen 2019* | Uganda | Bidibidi | Refugee, women | NR | 477 | 0.88 (0.84, 0.90) |
| **High Income Country** | | | | | | |
| Kim 2018 | South Korea | Unclear | Unmarried | NR | 161 | 0.06 (0.03, 0.10) |
| Moon 2020 | South Korea | Not clear | College | 2019 | 75 | 0.19 (0.11, 0.29) |
| Kuhlmann 2019 | USA | St Louis | Low-income | 2017 | 183 | 0.00 (0.00, 0.02) |
| Borowski 2011 | USA | Web-based | Colleagues | 2011 | 156 | 0.08 (0.05, 0.14) |
| Frank 2020 | USA | Midwest | Transgender | NR | 19 | 0.11 (0.01, 0.33) |
| Lamont 2019 | USA | Midwest | College | NR | 62 | 0.11 (0.05, 0.22) |

0   .25   .5   .75   1

**Fig 2. Use of reusable pads in surveys or at enrolment in a cohort, 2011–2019.** *According to the website, Bidibidi camp received reusable pads in August 2017 [57]. However, Lenia 2019 described that both reusable and disposable pads were distributed [53].

menstruation before the intervention. In a boarding school (secondary) in the same area over the same time period, uptake of RMPs ranged from 23–44% among schoolgirls of whom 93% reported to regularly use single-use pads before the intervention [12]. In several studies, participants reported the need for washing as the reason not to use RMPs and preferring single-use pads instead [42,44,54].

## Leakage using RMPs

The complaint of menstrual blood leakage among RMP-users was lower compared with a control group of cloth-users (44% vs. 78%, p<0.001) in one Ugandan study [24], and in two Ugandan cohort studies comparing RMPs use at endline against usual product at baseline (20% vs. 33%, respectively, [36]; 9% vs. 59%, [37], Table 3, no p-values provided). Fear of leaking was less or similar among RMP-users compared to cloth-users in a survey among schoolgirls in three states in India (10% vs. 20% respectively in Chhattisgarh, p = 0.08; 24% vs. 27% in

**Table 1B. Characteristics of cohort studies contributing to the Reusable Menstrual Pad review (alphabetical order).**

| | Studies | Material | Location (country & site) | Time | Sample size and population Prospective or retrospective | Information age participants | RMP brand (type) | Comparison (where applicable) | Follow up | Outcomes | Quality score |
|---|---|---|---|---|---|---|---|---|---|---|---|
| 1 | Bardsley 2020 [28] | Thesis | Thailand, Mae La refugee camp | 2019–2019 | 68 schoolgirls in boarding houses who had received reusable pads in last 18 months Retrospective | Range 13–22 yrs | Days for Girls kits (8) | Usual item | < = 18 months | Use of reusable pads, acceptability, education | 2 |
| 2 | Coker-Bolt 2017 [29] | Journal Article | Haiti, Leogane | 2016–2016 | 49 bachelor nursing students Prospective | Range 18–24 yrs | Days for Girls (8) † | Usual item, not specified | 2 months | Use of reusable pad, acceptability, school attendance | 3 |
| 3 | Femme International 2017 [30] | Report | Tanzania, Moshi | 2016–2017 | 233 schoolgirls (6 schools) and 100 women Prospective | Girls: mean 15.9 yrs, sd 1.2, n = 233 Women: mean 31.7, sd 9.0, n = 100 | AFRIpad, number not provided‡ | Usual item (80% disposable pads, 32% cloths), menstrual cups | 6–12 months | Use of reusable pad | 2 |
| 4 | Gade & Hytti 2017 [31] (Womena) | Report | Uganda, Rhino Camp Refugee settlement | 2017–2017 | 64 schoolgirls, 31 mothers/ guardians, 7 senior teachers or village workers Prospective | Mean 16 for girls, 25.9 for mothers/ guardians, no sd reported | AFRIpad (4)‖ received by 42 schoolgirls and 21 other women | Usual item (disposable pads, cloths, reusable pads, other) Menstrual cup | 4 months | Acceptability Feasibility Health and social impact in refugee context | 2 |
| 5 | Geismar 2018 [32] | Thesis | South Africa, location NR | 2017–2018 | 263 schoolgirls Retrospective | NR | Subz (6–9) ‖‖ | Usual item, not specified | ≤18 months | Use and acceptability of reusable pads | 2 |
| 6 | IFRC 2016 [33] Gilles-Hansen 2019 [34] IFRC 2013 [35] | Report | Locations in Uganda, Somaliland, Madagascar and Burundi | 2014–2015 | Women in selected communities Uganda 581 (refugee setting) Somaliland 371 Madagascar 720 Burundi 891 (refugee setting) Prospective | NR, range 12–50 years | AFRIpad or other (8)§ | Usual item (disposable pads, cloths, underwear) | 1 month 3 months | Use and acceptability of reusable pads | 3 |
| 7 | Kansiime 2020 [36] | Journal Article | Uganda, Entebbe | 2017–2018 | 232 schoolgirls Prospective | Mean 15.4 yrs, sd 1.3, n = 232 | AFRIpad (4)** | Usual item (not further specified) | 9 months | Use of reusable pad, acceptability, school attendance, psychosocial well-being | 3 |
| 8 | Kuncio 2018 [37] (UNHCR) | Report | Uganda, 3 refugee settlements | 2018–2018 | 237 schoolgirls in 3 refugee camps Prospective | 13–20 yrs | AFRIpad (4) ‖ | Disposable pads 71%, not further specified | 3 months | Acceptability of reusable pad | 3 |
| 9 | Mucherah 2017 [38] | Journal Article | Kenya, location unclear | 2014–2015 | 150 schoolgirls (51 from intervention and 99 from control school). Retrospective | Mean 13.1, sd 2.1, n = 150. Range: 11–16 | Brand not reported (4) | Usual item (not further reported); girls before menarche included | 12 months for RMP group | Acceptability, school attendance, psychosocial well-being | 2 |

*(Continued)*

**Table 1B.** (Continued)

| | Studies | Material | Location (country & site) | Time | Sample size and population Prospective or retrospective | Information age participants | RMP brand (type) | Comparison (where applicable) | Follow up | Outcomes | Quality score |
|---|---|---|---|---|---|---|---|---|---|---|---|
| 10 | Murthy 2015 [39] | Abstract | India, South Rajasthan | NR | 45, no further information | 16–45 yrs | Uger pads (number not reported) | Usual item (cloths and disposable pads) | 12–13 cycles | Acceptability | 2 |
| 11 | Nabata & Clayton 2019 [40] Hooper 2020 [41] | Abstracts | India, Spiti Valley | NR | 42 menstruating schoolgirls, boarding Retrospective | 14+ yrs | Brand not reported (4), menstrual cup received at same time†† | Usual item (not further reported) | 12 months 24 months | Preference of reusable pads or menstrual cups | 2 |
| 12 | Scott 2013 [12] | Report | Uganda, Mbale | 2014–2014 | 512 schoolgirls (primary and secondary) Prospective | NR | AFRIpad (5), KMET pad (6), Mwezi pad (4) Makapad (10) ‡‡ | Usual item: Cloths, disposable pads | 6 months | Use and acceptability of reusable pads | 3 |

NA, not applicable. NR, not reported. For details of brands mentioned in this table, see S1 File.

*Montgomery 2016: Two base pads, three attachable winged liners, three straight liners, and two small bags for carrying. Schoolgirls additionally received 3 pairs of underwear and one sachet of Omo (washing soap, 45 grams) [23].

†Coker-Bolt 2017 [29]: Days for Girls kit: Drawstring bag to contain content, 8 pads, two moisture barrier shields, instructions, one gallon-sized ziploc bag.

‡ Femme International 2017 [30]: Femme kit: Soap, towel, bowl and reusable pads, number not provided.

‖ Gade & Hytti 2017 [31] and Kuncio 2018 [37] (UNHCR): Afripad deluxe kit, containing 3 Maxi pads that can be worn 6–8 hours, a Super Maxi pad that can be worn 8–10 hours and a washable storage bag.

§Uganda: Rhino refugee camp received kit A with 16 disposable pads; Mungula camp received kit B with reusable pads (3 winged pads and 5 straight pads). Madagascar received kits A & B to all communes. Somaliland received kit C with 10 disposable pads and 1 pack of reusable pads, quantity not specified. Burundi received kits A (disposable pads) and B (reusable pads) (Bwagiriza refugee camp). All kits contained underwear (2), use, care and disposal instruction for item, polyethylene storage bag, plastic bucket with lid, bar of personal bathing soap. Kits with disposable pads also contained biodegradable plastic bags. Kits with reusable pads also contained plastic coated rope and pegs and laundry soap [33].

**Kansiime 2020 [36]: Menstrual management kit containing 4 AFRIpads, small towel, soap, water bottle, underwear, a mirror, and menstrual calendar.

††Nabata 2020 [40]: All participants received 1 menstrual cup, 4 reusable pads, cleaning supplies, and menstrual diary with training on usage and cleaning.

‡‡Scott 2013 [12]: Mwezi pads: Circular base with Velcro for attaching around the crotch of underwear, onto washable removable inserts are anchored; package with 4 inserts. KMET pads: Terrycloth with soil-resistant liner, locally-made: Package of 6. Afripad kit same as for Montgomery. All high school students in this study also received Makapads, locally made disposable sanitary pads, completely biodegradable except for plastic liner (required by Ugandan government).

‖‖ Geismar 2020 [32]: Subz contains 2–3 underwear and 6–9 reusable pads and educational booklet.

Maharashtra, p = 0.520; 10% vs. 16% in Tamil Nadu, p = 0.253) and similar or higher compared to single-use pad-users (10% vs. 12% respectively in Chhattisgarh, p = 0.774; 24% vs. 18% in Maharashtra, p = 0.044; 10% vs. 7% in Tamil Nadu, p = 0.288) [8]. In qualitative studies, some users mentioned that reusable pads or period underwear felt thin and were concerned it would cause leakage [22,33], whereas others reported they felt more secure against leaking with RMPs than single-use pads (S1 File) [28,53,58].

### Mobility, comfort, and odour using RMPs

Reduced mobility related to any type of menstrual product used was high (~40%) and was not significantly different when comparing RMPs with cloths or usual item used for menstruation (not further defined) in a Ugandan study among schoolgirls (Table 3) [24]. Single-use

**Table 1C. Characteristics of surveys contributing to the Reusable Menstrual Pad review (alphabetical order).**

| | Studies | Material | Location (country & site) | Time | Sample size and population | Information age participants | RMP brand (type) | Outcomes for systematic review | Quality score |
|---|---|---|---|---|---|---|---|---|---|
| 1 | Amulen 2019 [42]* | Thesis | Uganda, Bidibidi refugee camp | NR | 477 schoolgirls (60% primary) | Mean 17.1, no sd. Range 10–19 yrs | No particular brand | Use of reusable pads | 2 |
| 2 | Beksinska 2020 [43] | Journal Article | South Africa, Kwazulu Natal | 2017–2018 | 509 students, higher education | Mean 21, no sd | No particular brand | Use of reusable pads | 2 |
| 3 | Borowski 2011 [44] | Thesis | USA (web-based) | 2011–2011 | 155 women | Age ≥18 years, 43.4% 25–34 yrs | No particular brand | Use of reusable pads, consideration of reusable products | 2 |
| 4 | Crankshaw 2020 [45] | Journal Article | South Africa, Gauteng | 2018–2018 | 505 schoolgirls (secondary) | Median 17, IQR 16–18 | No particular brand | Use of reusable pads | 2 |
| 5 | Frank 2020 [46] | Journal Article | USA, Midwest | NR | 19 transsexual or binary persons | Range 18–29, mean 22, median 21 | No particular brand | Use of reusable pads | |
| 6 | Goodson 2020 [47] | Thesis | Kenya, Kibwezi | 2019–2019 | 50 menstruating schoolgirls | NR | No particular brand | Use of reusable pads | 2 |
| 7 | Hennegan 2020 [48] Hennegan 2020 [49] | Journal Article | Uganda, Soroti | 2019–2019 | 538 menstruating schoolgirls (12 schools) | 14.5, sd 1.2, n = 538 | No particular brand | Use of reusable pads | 2 |
| 8 | Kim 2018 [50] | Journal Article | South Korea, location unclear | Unclear | 161 unmarried women | 19–23 yrs: 55, 24–28 yrs: 53, 29–33 yrs: 36, 34–49 yrs: 17 | No particular brand | Use of reusable pads and satisfaction | 2 |
| 9 | Kuhlman 2019 [51] | Journal Article | USA, St Louis | 2017–2018 | 183 low-income women | 35.8, sd 13.3, n = 183. Range 18–69 | No particular brand | Use of reusable pads | 3 |
| 10 | Lamont 2019 [52] | Journal Article | USA, Louisville | NR | 62 undergraduate psychology students | Mean 20.3, sd 1.2, n = 62 | No particular brand | Use of reusable pads, willingness to use in future | 2 |
| 11 | Lenia 2019 [53]* | Thesis | Uganda, Bidibidi, refugee camp | NR | 422 women in refugee camp Retrospective | Mean 25, sd NR, Range: 15–49 yrs | No particular brand | Use of reusable pads, acceptability | 3 |
| 12 | Miiro 2019 [54] | Journal Article | Uganda, Entebbe | 2015–2016 | 352 schoolgirls | Mean 15.6, sd 1.1, n = 352. Range: 12–17 | No particular brand | Use of reusable pads, willingness to use in future, school attendance | 3 |
| 13 | Moon 2020 [55] | Journal Article | South Korea, location not clear | 2018–2019 | 75 unmarried university students | Mean 23.2, sd 1.9 | No particular brand | Use of reusable pads | |
| 14 | Sivakami 2018 [8] | Journal Article | India: Tamil Nadu Chhattisgarh Maharashtra | 2015–2016 | 2564 menstruating schoolgirls | Mean 14.1, sd 1.1, n = 2533 | No particular brand | Use of reusable pads, leakage, mobility, school attendance | 2 |
| 15 | SNV 2014 [56] | Report | Uganda: Dokolo, Lira, Mubenda, Mukono | 2014–2014 | 2609 schoolgirls (606 schools) | NR | No particular brand | Use of reusable pads | 2 |

NA, not applicable. NR, not reported. For details of brands mentioned in this table, see S1 File.

* Bidibidi camp received reusable pads in August 2017 [57]. However, Lenia (2019) [53] describes that both reusable and disposable pads were distributed.

compostable pads contributed significantly more to overall wellbeing than RMPs in a study in slums in India [27]. In a Ugandan study of menstrual cups and RMPs, 68% of RMP-users were satisfied with being able to do activities, compared to 88% of menstrual cup users [31]. There were complaints among RMP-users that the RMPs were too big [29,31] or did not stay in

**Table 2. Uptake of reusable menstrual pads in cohort studies.**

| Study | Country, population | RMP brand (number received) | Use of RMP at baseline | Instruction | No of recipients of RMP at baseline | Follow-up time (months) | Use at follow up or endline, % (n/N) | Preference or reason for not using RMP |
|---|---|---|---|---|---|---|---|---|
| Bardsley 2020 [28] | Thailand, schoolgirls, refugee setting | Days for Girls kits (8) | NR | NR | 68 | 18 | 94.1% (64/68) | "Students said that compared to disposables, the DfG pads are less itchy, cooler to wear and feel comfortable because they are more secure" |
| Coker-Bolt 2017 [29] | Haiti, students | Days for Girls (8) | NR | Verbally by local producer | 49 | 2 | 89.8% (44/49) | NR |
| Femme International, 2017 [30] | Tanzania: schoolgirls | AFRIpad (number NR) | 0% | Interactive workshops, 2 hrs/day for 4 days | 459 RMP recipients, 110 surveyed at 6 m, 13 at 12 months | 6 12 | 95.7% (105/110, sample from 5 schools) 92.3% (12/13 after 1 year, sample in 1 school) | • No underwear or trouble attaching pad • Itching/chafing • No access to water, soap, inability to dry in sunlight (Numbers not provided) 65% of menstrual cup recipients used at 6 months, 78.6% at 1 year |
| Femme International, 2017 [30] | Tanzania: women (orphanage) | AFRIpad (number NR) | NR | Interactive workshops, 2 hrs/day for 4 days | 40 RMP recipients, 4 surveyed at 6m | 6 | 100% (4/4) | NR 28.6% (2/7) of menstrual cup recipient used at 1 year |
| Gade & Hytti 2017 [31] | Uganda, refugee camp, women | AFRIpad (4) | 8.3% (5/63) | 3-hour workshop | 63 RMP recipients (42 schoolgirls & 21 women) | 4 | • 100% of reusable pad recipients used reusable pad during last menstruation, N not clear | 69% liked RMPs Problem areas: • Lack of underwear (8%) • Uncomfortable (12%) • washing/changing difficult (19%) 61% of menstrual cup recipients used cup during last menstruation |
| Garikipati 2019 [27] | India, slums | Safepad (4) | 0% (0/293) | Local research assistant | 133 | 6 | 125/125 (100) 20.8% (26/125) in combination with other methods | NR |
| Geismar 2020 [32] | South Africa, Durban, schoolgirls | Subz (4) | NR | Workshop | 263 (Retrospective cohort) | 6 | 51.0% (134/263) | Pad too big or too small, too bulky, leaks through, don't like to wash pad with blood, prefers disposable pads, tends to slide/move with activity |
| Hennegan 2016 [24] | Uganda, schoolgirls | Afripad (6) | NR | Locally trained research assistants | 87 | 12–24 | 82.8% (72/87) | • Preferred disposable pads (2) • "Felt the reusable pad burned" (1) • "Did not look like they would work well" (1) |
| IFRC 2016 [33]* | Uganda, refugee camp, women | AFRI pad or another reusable pad (8) | NR | Education and demonstration sessions by volunteers | 791 kits distributed, 318 RMP recipients for survey | 3 | 100% (318/318) | • 56% preferred disposable pads at 3 months, but information not split by type of item they received at baseline 12–17 yrs: 86% 18–34 yrs: 54% 35–50 yrs: 41% • 49% of RMP receivers: RMP comfortable to use (1 month follow up). |
| IFRC 2016 [33]* | Somaliland, women | Afripad or another reusable pad (NR, 10 disposable pads in same kit) | NR | Leaflets and community information | 371 RMP recipients (in same kit also 10 disposable pads) | 3 | 63% (233/371), only 2% (7) used exclusive RMP | • 32% preferred disposable pads at 3 months 12–17 yrs: 37% 18–34 yrs: 33% 35–50 yrs: 21% |
| IFRC 2016 [33]* | Madagascar, women | AFRI pad or another reusable pad (8) | NR | Not clear | 1000 kits distributed, 360 RMP recipients for survey | 3 | No data available | • 40% preferred disposable pads at 3 months, but information not split by type of item they received at baseline (not available by age group). |

*(Continued)*

**Table 2.** (Continued)

| Study | Country, population | RMP brand (number received) | Use of RMP at baseline | Instruction | No of recipients of RMP at baseline | Follow-up time (months) | Use at follow up or endline, % (n/N) | Preference or reason for not using RMP |
|---|---|---|---|---|---|---|---|---|
| Giles 2019 [34]* IFRC 2013 [35] | Burundi, women | AFRI pad or another reusable pad (8) | NR | Not clear | 2000 kits distributed, 891 RMP recipients for survey | 1 3 | No data available | "Women and girls preferred the reusable kits more than the disposable kits." Adolescent girls: 90% satisfied with kit B (reusable), 68% satisfied with kit A (disposable). Women (35–50 yrs): 85% satisfied with kit B, 65% satisfied with kit A. |
| Kansiime 2020 [36] | Uganda, schoolgirls | AFRIpad (4) | 18.5% (41/222) | NGO staff (Womena*) | 222 | 9 | 82.5% (155/188) | NR |
| Kuncio 2018 [37] (UNHCR) | Uganda, refugee camps, schoolgirls | AFRIpad (4) | NR | 3 hr training by AFRIpad staff or staff trained by AFRIpad | 168 schoolgirls in 2 camps | 3 | • 99% tried (166/168) • 92% (155/168) used during last period, • 79% (133/168) used it as main method • 99% intends to continue using Afripad | • 97% recommend to friend • 83.6% preferred AFRIpads over disposable pads • 7.9% liked combination with disposable pads • 8.5% preferred disposable only Reasons for not using: light flow (1), in exams and no time to wash, itching/burning (1) |
| Mucherah 2017 [38] | Kenya, schoolgirls | Brand NR (4) | NR | Workshop on reusable pads | 51 | 12 | 66.7% (34/51) | NR |
| Murthy 2015 [39] | India, rural women | Uger (NR) | 0% (9 cloth, 36 disposable pads) | Not reported | 45 | 12–13 cycles | 12m: 100% (45/45) | NA |
| Nabata & Clayton 2020 [40] | India, boarding schoolgirls | NR (4), received cup at same time | NR | Health workshop | 42 | 12 24 | 12m: 80% preferred reusable pad (16/20) 24m: 43% (9/20) | • 12m: 30% (6/20) preferred menstrual cup • 24m: 10% (2/20) preferred cup Reasons for not using cup: wearing down, lost |
| Scott 2013 [12] | Uganda, primary schoolgirls | AFRIpad (5) Handmade cloth (MWEZI, 4) | 0% | Head mistress of schools | 55 Afripad 55 Mwezi | 6 | 100% intended to continue using assigned pad | • Most girls used cotton wool for menstruation before intervention |
| Scott 2013 [12] | Uganda, secondary schoolgirls | KMET (6) AFRIpad (5) Handmade cloth (MWEZI pad, 4) | NR | Community women | KMET 134 Afripad 134 Mwezi 134 All girls also received disposable Makapads (see Table 1) | 6 | Intends to continue using: 25.8% of KMET recipients (34/134) 43.8% of Afripad recipients (59/134) 22.6% of MWEZI recipients (30/134) | • 34.6% of all girls (139/402) preferred disposables after 6 months ("disposables better than reusable pads") • Number of girls continuing buying disposable pads during follow up time: ○ 52.0% of KMET users ○ 53.4% of Afripad users ○ 73.0% of MWEZI users • "about half of all girls will not switch from disposables even if provided with a good cloth alternative for free." |

*Uganda: Rhino refugee camp received kit A with 16 disposable pads; Mungula camp (Uganda) received kit B with reusable pads (3 winged pads and 5 straight pads). Madagascar received kits A & B for all communities. Somaliland received kit C with 10 disposable pads and 1 pack of reusable pads, quantity not specified. Burundi received kits A (disposable pads) and B (reusable pads) (Bwagiriza refugee camp). All kits contained underwear (2), use, care and disposal instruction for item, polyethylene storage bag, plastic bucket with lid, bar of personal bathing soap. Kits with disposable pads also contained biodegradable plastic bags. Kits with reusable pads also contained plastic coated rope and pegs and laundry soap.

place: "they run away when we are playing" [59]. In India, discomfort when moving or sitting was not significantly different among RMP-users, cloth-users, or single-use pad users [8]. Worrying about odour (57%) or experiencing odour (29%) among RMP-users in Uganda was

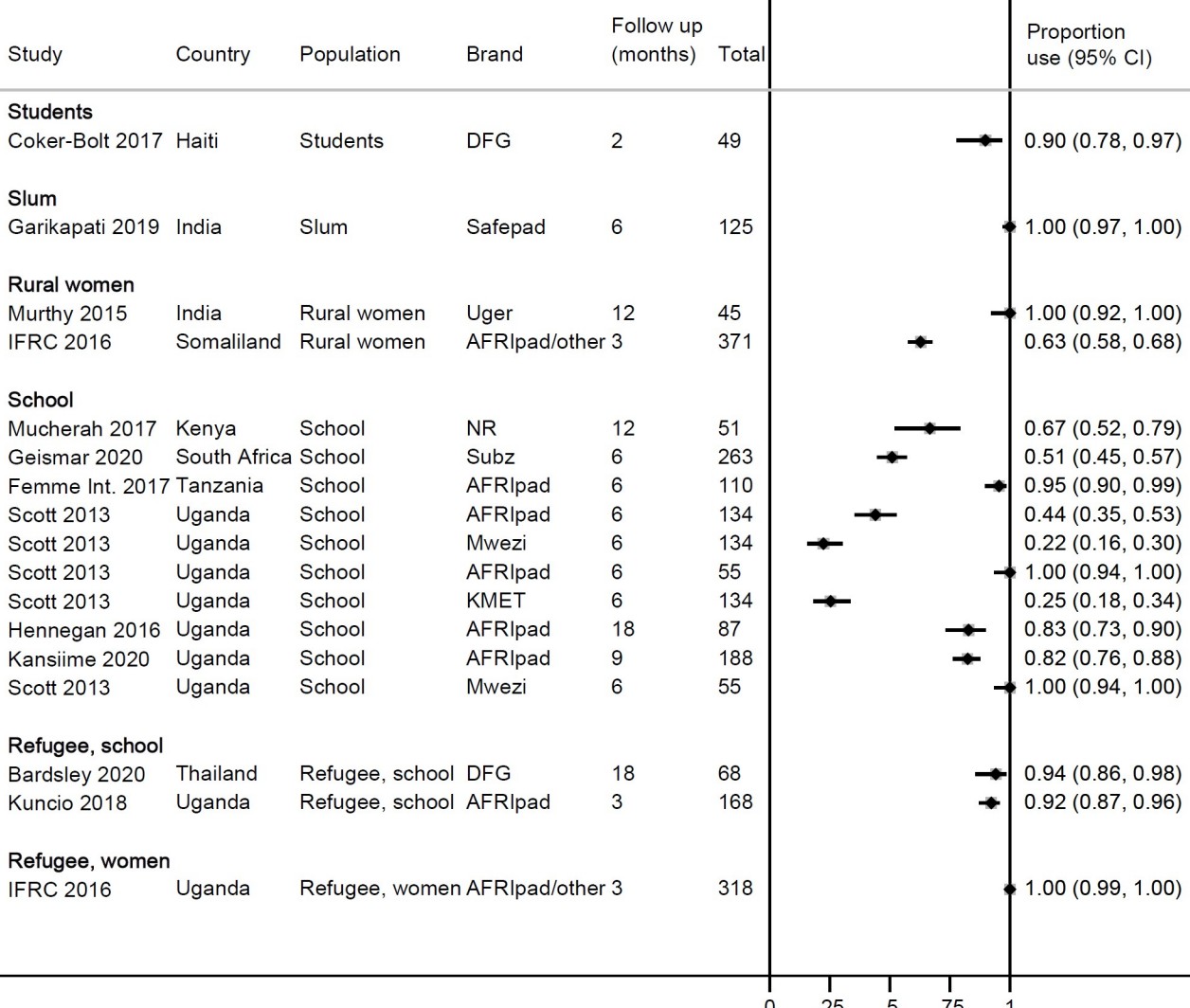

| Study | Country | Population | Brand | Follow up (months) | Total | | Proportion use (95% CI) |
|---|---|---|---|---|---|---|---|
| **Students** | | | | | | | |
| Coker-Bolt 2017 | Haiti | Students | DFG | 2 | 49 | | 0.90 (0.78, 0.97) |
| **Slum** | | | | | | | |
| Garikapati 2019 | India | Slum | Safepad | 6 | 125 | | 1.00 (0.97, 1.00) |
| **Rural women** | | | | | | | |
| Murthy 2015 | India | Rural women | Uger | 12 | 45 | | 1.00 (0.92, 1.00) |
| IFRC 2016 | Somaliland | Rural women | AFRIpad/other | 3 | 371 | | 0.63 (0.58, 0.68) |
| **School** | | | | | | | |
| Mucherah 2017 | Kenya | School | NR | 12 | 51 | | 0.67 (0.52, 0.79) |
| Geismar 2020 | South Africa | School | Subz | 6 | 263 | | 0.51 (0.45, 0.57) |
| Femme Int. 2017 | Tanzania | School | AFRIpad | 6 | 110 | | 0.95 (0.90, 0.99) |
| Scott 2013 | Uganda | School | AFRIpad | 6 | 134 | | 0.44 (0.35, 0.53) |
| Scott 2013 | Uganda | School | Mwezi | 6 | 134 | | 0.22 (0.16, 0.30) |
| Scott 2013 | Uganda | School | AFRIpad | 6 | 55 | | 1.00 (0.94, 1.00) |
| Scott 2013 | Uganda | School | KMET | 6 | 134 | | 0.25 (0.18, 0.34) |
| Hennegan 2016 | Uganda | School | AFRIpad | 18 | 87 | | 0.83 (0.73, 0.90) |
| Kansiime 2020 | Uganda | School | AFRIpad | 9 | 188 | | 0.82 (0.76, 0.88) |
| Scott 2013 | Uganda | School | Mwezi | 6 | 55 | | 1.00 (0.94, 1.00) |
| **Refugee, school** | | | | | | | |
| Bardsley 2020 | Thailand | Refugee, school | DFG | 18 | 68 | | 0.94 (0.86, 0.98) |
| Kuncio 2018 | Uganda | Refugee, school | AFRIpad | 3 | 168 | | 0.92 (0.87, 0.96) |
| **Refugee, women** | | | | | | | |
| IFRC 2016 | Uganda | Refugee, women | AFRIpad/other | 3 | 318 | | 1.00 (0.99, 1.00) |

0    .25    .5    .75    1

**Fig 3. Uptake of reusable pads in cohort studies in middle- and low-income countries, 2014–2019.** DFG: Days for Girls. IFRC: International Federation of Red Cross and Red Crescent Societies.

not significantly different from cloth-users (60% and 33%, p = 0.736 and p = 0.632 respectively) [24]. In another Ugandan cohort, 77% of RMP-using schoolgirls were satisfied with the absence of smell compared with 88% among the menstrual cup users in the same study [31] (Table 3).

## Washing, drying, and changing RMPs

In five studies among schoolgirls and students in three countries, 44–91% (median 80%) of participants thought the RMPs were easy to clean (Table 4) [28,29,31,32,37]. In three studies (four locations, two in refugee camps in Uganda), a median of 16% of participants (range 6–27%) reported they had difficulty in finding enough water for washing the RMPs (Table 4) [31,33,37]. This was also reported for period underwear in a qualitative study (S8 Table in S1 File) [22]. In two quantitative and five qualitative studies (all in LMICs), participants reported feeling disgust at having to wash menstrual blood (range 3–22%) [24,31,32,37,42,47,60]. Most

**Table 3.** Information on leaking, mobility and odour when using reusable menstrual pad.

| Study | Country, study design, population, follow up time | RMP brand (number provided) & Alternative | Leakage | | Mobility/comfort | | Odour | |
|---|---|---|---|---|---|---|---|---|
| | | | RMP or endline | Alternative or baseline | RMP or endline | Alternative or baseline | RMP | Alternative or baseline |
| Hennegan 2016 [24] RH | Uganda, endsurvey trial, 205 schoolgirls, 24 months | AFRIpad (6) vs. control group of cloth-users or users of other items for menstruation; disposable pad-users (16) were excluded from control group. | Afripads: Leakage a problem: 44.4% (32/72, p<0.001) Outside garment soiling: 43.6% (24/55, p = 0.990) | Control group: Leakage a problem: 78.4% (87/111) Outside garment soiling: 43.5% (37/85) | Afripads: 47.3% (26/55) feared that pad would fall out (p = 0.805) 37.5% (24/64) avoided exercise (p = 0.419) 37.5% (24/64) unable to play (p = 0.571) | Control group: 49.4% (42/85) feared that pad would fall out 43.8% (46/105) avoided exercise 41.9% (44/105) unable to play | Afripads: 29.1% (16/55) experienced bad odour (p = 0.632) 56.9% (41/72) worried about odour (p = 0.736) | Control group: 32.9% (28/85) experienced bad odour 59.5% (66/111) worried about odour |
| Garikipati 2019 [27] | India, trial, slums | Safepad vs. single use compostable item vs. education only | No information | No information | RMP: Convenient to use 54.4% (68/125) (p<0.0001) Contributed to overall wellbeing 39.2% (49/125) (p = 0.022) | Single use compostable item: Convenient to use: 85.8% (109/127) Contributed to overall wellbeing 54.3% (69/127) | No information | No information |
| Bardsley 2020 [28] | Thailand, cohort, schoolgirls, 18 months, refugee setting | Days for Girls (8) vs. usual item | No information | No information | 91.2% (62/68) comfortable to wear | NR | No information | No information |
| Coker-Bolt 2017 [29] | Haiti, cohort, students, 2 months | Days for Girls (8) vs. usual item | No information | No information | 84.1% (37/44) able to participate in all daily activities with RMP. 11.4% had problems (5/44): Liner not always secure in underwear; "pads are too big" | NR | No information | No information |
| Gade & Hytti 2017 [31] | Uganda, cohort, refugee camp, schoolgirls and women, 4 months | AFRIpad (4) vs Ruby menstrual cup | As benefits of reusable pad compared to usual method, "no leaking" mentioned by ~30%, no denominator available | NR | Endline: As benefits of reusable pad compared to usual method, "comfort and freedom to play" mentioned by ~40%. 68% very satisfied with being able to do normal activities. 12% uncomfortable: pads not staying in place, too big. No denominator available. | Endline MC: 88% very satisfied with being able to do normal activities when using MC, compared to usual item. No denominator available. | As benefits of reusable pad compared to usual method, "no smell" mentioned by ~55%. 77% very satisfied with absence of smell. No denominator available | Endline MC: 88% very satisfied with absence of smell when using MC, compared to usual item. No denominator available. |
| Geismar 2018 [32] | South Africa, cohort, schoolgirls, 6 months | Subz (4) | 27.0% (71/263) pad absorbent for 3–6 hrs | Not applicable | 65.0% (171/263) pad attaches easily to panty 36.8% (97/263) pad comfortable to wear | Not applicable | No information | Not applicable |
| Kansiime 2020 [36] | Uganda, cohort, schoolgirls, 9 months | AFRIpad (4) at endline vs usual item at baseline | Endline with 82.5% RMP use: leakage 19.7% (36/183) Underwear stained: 27.3% (50/183) | Baseline with 18.5% RMP use: leakage 33.3% (74/222) Underwear stained: 23.4% (52/222) | No information | No information | No information | No information |
| Kuncio 2018 [37] UNHCR | Uganda, cohort, schoolgirls, refugee camps, 3 months | Afripad (4) at endline vs. usual item at baseline | Endline: 9.2% leaks (15/167) | Baseline: 58.5% leaks (146/249) | Endline: 88.6% satisfied with ability to continue doing normal activities | | No information | No information |
| Murthy 2015 [39] | India, cohort, rural, 12 months | Uger pads (NR) | No information | No information | 40.0% (18/45) no discomfort when wearing | No information | No information | No information |

*(Continued)*

**Table 3.** (Continued)

| Study | Country, study design, population, follow up time | RMP brand (number provided) & Alternative | Leakage | | Mobility/comfort | | Odour | |
|---|---|---|---|---|---|---|---|---|
| | | | RMP or endline | Alternative or baseline | RMP or endline | Alternative or baseline | RMP | Alternative or baseline |
| Nabata 2019 [40] Hooper 2020 [41] | India, cohort, schoolgirls (boarding), 12–24 m | RMP (brand NR) vs. menstrual cup | 21.2% leakage in RMPs Numbers not reported (Likely very small sample: at start N = 42) | 66.7% leakage in cups Numbers not reported (Likely very small sample: at start N = 42) | 16.7% pain and discomfort Numbers not reported | 60.0% pain and discomfort Numbers not reported | No information | No information |
| Sivakami 2019 [8] | India, Chhattisgarh, survey, schoolgirls | RMP (brand NR) vs cloths & disposable pads | Fear of staining: RMP 10.3% (6/58, p = 0.080 vs. cloths, p = 0.774 vs. disposable pads) | Fear of staining: Cloths 19.8% (112/566), Disposable pads 11.7% (27/231) | Discomfort when moving/sitting: RMP 5.2% (3/58, p = 0.242 vs. cloths, p = 0.630 vs. disposable pads) | Discomfort when moving/sitting: Cloths 9.9% (56/566) Disposable pads 6.9% (16/231) | No information | No information |
| | India, Maharashtra, survey, schoolgirls | RMP (brand NR) vs cloths & disposable pads | Fear of staining: RMP 23.7% (73/307, p = 0.520 vs. cloths, p = 0.044 vs. disposable pads) | Fear of staining: Cloths 26.9% (28/104) Disposable pads 17.6% (69/392) | Discomfort when moving/sitting: RMP 3.6% (11/307, p = 0.403 vs. cloths, p = 0.701 vs. disposable pads) | Discomfort when moving/sitting: Cloths 1.9% (2/104) Disposable pads 3.1% (12/392) | No information | No information |
| | India, Tamil Nadu, survey, schoolgirls | RMP (brand NR) vs cloths & disposable pads | Fear of staining: RMP 9.5% (16/169, p = 0.253 vs. cloths, p = 0.288 vs. disposable pads) | Fear of staining: Cloths 15.8% (6/38) Disposable pads 7.0% (37/530) | Discomfort when moving/sitting: RMP 0.6% (1/169, p = 0.243 vs. cloths, p = 0.061 vs. disposable pads) | Discomfort when moving/sitting: Cloths 2.6% (1/38) Disposable pads 3.2% (17/530) | No information | No information |
| Scott 2013 [12] | Uganda, cohort, primary schoolgirls, 6 months | AFRIpad (5) vs. handmade reusable pad (MWEZI, 4) | MWEZI pads: inserts too long or too short, still extra layers added because of leaking AFRIpad: no extra layers needed (no numbers provided) | NA | Can take part in sports when using RMP (no numbers provided) | NA | MWEZI (hand-made) pads were beginning to smell after 6 months (no numbers provided) | NA |

Abbreviation: MC, menstrual cup. RMP, reusable menstrual pad.

*Disposable pad-users (16) were excluded from this group. The control group included users of cloths and other items.

participants in two studies used soap when washing (range 65–95%) [24,31,37]. Lack of equipment such as soap or a bucket, or problems with finding a private place for washing [31,33,37], drying [33] or changing RMPs [58] were reported. However, a study in Thailand noted that washing and drying RMPs was easier than finding places to dispose of used single-use pads in a refugee camp setting [28]. Although drying outside in the sun is recommended for RMPs [61], there was reluctance as others might see the RMPs, thus some participants reported hiding it under another piece of laundry [53,62]. Some participants complained RMPs required long drying times [24,31,59] of 4 hours to two days, which could result in wearing them while still damp (range 10–14% reported in two studies) [24,37]. This problem was exacerbated by the rainy season and the low number of RMPs available per menstruation [31,37]. Schoolgirls reported problems such as lack of privacy for changing at school (Hennegan et al 2016: 25% among RMP-users, 42% among cloth users, p = 0.017) [24,31,37]; some avoided changing because they did not want to carry the used RMP around [58]. In a Ugandan study, RMP-users were more likely to dry the reusable pad outside compared to cloth-users [24]. The study

reported that the RMPs dried faster than cloth, and users were less likely to wear damp RMPs compared to cloth-users [24]. RMP-users were more likely to change three or more times per day compared to usual practice product, a potentially more hygienic habit, but the reason for frequency of change was not clear (e.g., hygiene, education, or lower absorbency of RMP, Table 4) [24,25]. Time constraints to wash RMPs were a reason not to use them [22,47,60]. In the USA, homeless women did not consider RMPs to be practical because of the difficulty in cleaning due to issues of mobility (constantly moving around the houses of friends and hostels) and lack of privacy in shared cleaning facilities [63].

## Safety of RMPs

We intended to evaluate serious adverse events, and effects on perineal skin, and infections of reproductive or urinary tract infections in association with RMP-use. No adverse events related to RMPs were identified in the Manufacturer and User Facility Device Experience Database (MAUDE) maintained by the US Food and Drug Administration (S1 File). None of the studies used an objective measure to assess safety of RMPs and all complaints were based on self-report. Complaints of itching, burning or chaffing were noted by two out of 13 (15%) girls after using a RMP for one year (no baseline information or control group available, Table 5) [30], and among 40 out of 267 (24%) schoolgirls in a refugee camp after three months of RMP use [37]. In comparison, in the latter study, 73% reported itching or burning when using single-use pads at baseline, with 20% reporting they had reused single-use pads because of lack of resources [37]. Although studies by the International Red Cross in refugee camps noted self-reported complaints on itching, burning and infections, they distributed menstrual kits with RMPs and single-use pads, combined or separately, and did not report complaints by type of kit. These self-reported complaints ranged from 0.3–21% at 1–3 months post-distribution, compared to 19–27% at baseline [33]. Some studies did not report the percentage of complaints, but noted that these complaints were associated with wearing the same RMP for an extended duration [26] or with inadequate cleaning or drying [42,60]. A study in Malawi suggested that the materials used to make RMPs, such as cheap cottons, could cause skin irritation and make it hard to walk, especially if the RMP was still damp [60]. A small Indian study (~20 at follow up 12–24 months) among boarding-school girls who received both RMPs and menstrual cups noted a lower percentage of pain and discomfort when RMPs were used (17%) than when menstrual cups were used (60%). Two Ugandan studies compared inadequate menstrual practices among RMP-users and single-use pad and cloth-users (adequate menstrual practices: access to clean absorbents, adequate frequency of changing of the absorbent, washing of the body with water and soap, adequate disposal and privacy for managing menstruation) [25,53]: in a refugee camp, adequate menstrual hygiene management practices were 50% among RMP-users compared to 65% among single-use pad users and 78% among cloth-users [53]. In a school-based Ugandan study, adequate menstrual hygiene management practices were 11% among RMP-users compared to 9% among users of other materials [25] (Table 5). Two study participants in Argentina noted that allergies associated with single-pad use resolved when they swapped to an RMP [64]. No reports on severe or life-threatening adverse events were identified. Sharing of RMPs was reported by 6.7% (21/352) participants in a Ugandan school survey [54]. The effects of RMPs on school attendance were inconsistent (reported in the S1 File). New types of RMPs are still being developed (reported in S1 File).

## Product visibility of RMPs and costs

On 80 websites with educational materials on puberty and menarche, RMPs were mentioned as an option by 31 (39%), single-use pads by 61 (76%), tampons by 49 (61%), and menstrual

**Table 4. Information on washing, drying and changing of reusable menstrual pad.**

| Study | Country, study design, population, follow up time | Brand of RMP (number) & Alternative | Washing | | Drying | | Changing | |
|---|---|---|---|---|---|---|---|---|
| | | | RMP | Alternative or baseline | RMP | Alternative or baseline | RMP | Alternative or baseline |
| Amulen 2019 [42] | Uganda, survey, schoolgirls, refugee camp | Not reported | No information | No information | 74.4% (311/418) does not think the pad should be dried in a hidden place | No information | No information | No information |
| Hennegan 2016 [24,25] | Uganda, endsurvey trial, 205 schoolgirls, 24 months | AFRIpad (6) vs. cloths & disposable pads | Disgusted to wash absorbent: 22% (16/72, p = **0.048**) Washed using soap: 65% (47/72, p<**0.001**) | Disgusted to wash absorbent 36% (40/111) Washed using soap: 19% (24/129) | Dried outside 29% (21/72, p = **0.001**) <2 hours to dry absorbent 26% (18/72, p = 0.117) Wears damp pads: 14% (10/72, p<**0.001**) | Dried outside 10% (13/129) <2 hours to dry absorbent 36% (40/111) Wears damp pads: 48% (53/111) | 67% (48/72, p<**0.001**): Change 3 times or more 25% (18/72): Problem to change at school (p = **0.017** vs. cloths, p = 0.205 vs. disposable pads) | 19% (25/129): Change 3 times or more Cloth 42% (47/111), Disposable pads 11% (2/18): Problem to change at school |
| Bardsley 2020 [28] | Thailand, survey, schoolgirls, refugee camp | Days for girls (8) vs usual item | Easy to clean: 80.9% (55/68) | No information | No lack of facilities for cleaning/drying pads | No information | No information | No information |
| Coker-Bolt 2017 [29] | Haiti, cohort, students, 2 months | Days for girls (8) vs usual item | Easy to clean: 79.5% (35/44) Hard to clean: 13.6% (6/44) | No information | 9.1% (4/44) recommended extra pads due to long drying time | No information | No information | No information |
| Gade & Hytti 2017 [31] | Uganda, cohort, refugee camp, schoolgirls and women, 4 months | AFRIpad (4) vs Ruby Menstrual cup | RMP (no denominator available): • 20% hard to find water for washing • 49% hard to get soap • 3% hard to touch my blood • 81% had no separate washing basin • 15% washing difficult (takes time, privacy) • 65% satisfied with ease of cleaning | MC (no denominator available): • 4% difficult to clean Other problems: finding a container to clean • 77% satisfied with ease of cleaning ~70% boils in water | RMP (no denominator): • 20% hard to dry "In rainy season easy to wash but hard to dry" | No information | RMP (no denominator available): • 4% changing is difficult • 96% happy changing at home • 53% happy changing at school "Pads smelling in storage bag" | MC (no denominator available): • 92% happy changing at home • 48% happy changing at school |
| Geismar 2018 [32] | South Africa, cohort, schoolgirls, 6 months | Subz vs. usual item | 44.1% (116/263) easy to clean | No information | 44.1% (116/263) easy to dry | No information | No information | No information |
| IFRC 2016 [33] | Uganda, cohort, women, 3 months, refugee camp | RMP (AFRIpad, NR) or disposable pads in kits | One-month post distribution: 56% of RMP users washed pads in bathing areas 3 months post distribution: 7% problems finding a private place for washing, 6% problems for enough water | No information | One-month post distribution: 25% of RMP users dried pads inside house, 33% in bathing areas 11% no private place to dry | No information | No information | No information |
| | Somaliland, cohort, women, 3 months | RMP and disposable pads in kits | 3 months post distribution: 3% problems finding a private place for washing, 12% problems for enough water | | 3 months post distribution: 12% no private place to dry | | 3 months post distribution: 3% no private place to change | |
| | Madagascar, cohort, women, 3 months | RMP (AFRIpad, NR) or disposable pads in kits | "Main challenges lack of water for washing/ hygiene" Numbers not provided | No information | No information | No information | "Main challenges difficulty in finding a private area to change and dry pads" Numbers not provided | No information |

(*Continued*)

**Table 4.** (Continued)

| Study | Country, study design, population, follow up time | Brand of RMP (number) & Alternative | Washing | | Drying | | Changing | |
|---|---|---|---|---|---|---|---|---|
| | | | RMP | Alternative or baseline | RMP | Alternative or baseline | RMP | Alternative or baseline |
| Kuncio 2018 [37] UNHCR | Uganda, cohort, schoolgirls, refugee camps, 3 months | Afripad (4) at endline vs. usual item at baseline | Endline: • 26.8% not enough water for cleaning during menstruation. • 95% washed RMP in soap and cold water • 73.3% comfortable with washing blood from pads • 41.3% comfortable with washing in front of others • 90.6% satisfied with ease of washing | Baseline: 35.1% not enough water for cleaning during menstruation | • 69% dried RPM at clothesline outside. • 40% reported >4 hrs drying in wet season (vs. 1–2 hrs in dry season) • 12%: RPM never dried when raining. • 90% said they never wore a damp pad (4-pack considered insufficient: 6 or 8 better) • 82.0% comfortable with drying pads | No information | • 80% changed AFRIpad ≥ 2x/day, mainly in school. • 79.9% no problems with changing RPM in school • 94.7% no problems with changing RPM at home | No information |
| Lenia 2019 [53] | Uganda, survey, women, refugee camp | RMP (NR) vs. usual item | No information | No information | 4.3% (53/1243) Dried absorbent in the sun 86% (37/53) Dried in the sun but under another cloth | No information | No information | No information |
| Nabata 2019 [40] Hooper 2020 [41] | India, cohort, schoolgirls (boarding), 12–24 m | RMP (brand NR) vs. menstrual cup | Difficulty cleaning: RMP 57.9% | Difficulty cleaning: Cup 0.0% | No information | No information | No information | No information |
| Scott 2013 [12] | Uganda, cohort, primary schoolgirls, 6 months | AFRIpad (5) vs. handmade reusable pad (MWEZI, 4) | No information | No information | AFRIpads: 4 hrs-2 days drying time | MWEZI pads: 1–3 days drying time | No information | No information |

*No denominator for Pad users.

**No denominator provided.

cups by 33 (41%) (S1 File). We identified at least 110 brands of RMPs but could only access 73 websites (17 countries). Thirteen were in LMIC and 60 in HIC countries (for a summary of the findings see S1 File). For 69 RMP brands, prices were obtained for one single average product (e.g., daytime pad for regular bleeding, shipping costs not included, S13 Table in S1 File). The mean price per RMP was $8.95 (sd $5.08, range $1.00–21.96, median $8.33, n = 69). In LMIC this was $2.06 (sd $0.99, range $1.00–3.75, median $1.65, n = 10) and in HIC $10.11 (sd $4.54, range $2.17–21.96, median $9.75, n = 59). An estimate of the lifespan of RMPs was found for 34 brands with a mean of 4.3 years (sd 2.3, median of 4 years, range 1–10 years). The mean estimated lifespan for brands in LMIC was 2.9 years (sd 1.4, median 3, range 1–5 years, n = 11) and less than the mean lifespan in HIC (mean 4.9, sd 2.4, median 4, range 2–10 years, n = 23). In the included studies in LMIC, girls or women were provided 4–9 RMPs, often as part of a menstrual kit. *Afripad*, used in 9 studies, had an estimated lifespan of one year and kits contained 4–8 RMPs of 2–3 different sizes, with a cost estimate of 4–6 US$ per kit (S4 Table in S1 File). *Days for Girls*, used in 2 studies, had an estimated life span of 3 years with a kit containing two shields and 8 liners; the price depends on region, but is estimated at 11–17 US$ in East Africa (Kenya, Uganda, Malawi) (S4 Table in S1 File). The kits used in these studies were donated; it is not clear what the prices and availability are for local schoolgirls. The implicit assumption is that girls wash the RMPs during their menstruation, because the number of pads within the kit would not be sufficient to cover a menstruation of 5 days with 8-hourly changes. RMP producers in HIC recommended a higher number (and different

**Table 5. Any information that may relate to adverse effects of reusable menstrual pads.**

| Study | Country, study design, population, follow up time, sample at enrolment | RMP brand (number given) | RMP or endline | Alternative or baseline |
|---|---|---|---|---|
| Femme international 2017 [30] | Tanzania, cohort, schoolgirls, 6–12 months, N = 233 | AFRIpad (number NR) | 15% (2/13) girls reported 'itching and chaffing' with RMP after 1 year | No baseline information |
| IFRC 2016 [33]* | Uganda (refugee setting), women, cohort, 1–3 months, N = 581 | AFRI pad or another reusable pad (8) | Complaints of irritation or itching at 1 month: 9.8%. At 3 months: 21%. Kits with RMPs (Kit B) and single-use pads (Kit A) were distributed. Results were not stratified by kit received | Baseline: 24.0% complaints of irritation or itching during last menstruation with usual item. Denominator not reported |
| IFRC 2016 [33]* | Somaliland, women, cohort, 1–3 months, N = 371 | Afripad or other reusable pad (NR, 10 disposable pads in same kit) | Complaints of irritation or itching or smelly discharge at 1 month: 0.3%. At 3 months NR. RMPs and single-use pads were distributed in the same kit (Kit C). | Baseline: 19.4% complaints of irritation or itching or smelly discharge during last menstruation with usual item. Denominator not reported |
| IFRC 2016 [33]* | Madagascar, women, cohort, 1–3 months, N = 720 | AFRI pad or another reusable pad (8) | Complaints of infections and itching at 3 months: 10%. Kits with RMPs (Kit B) and single-use pads (Kit A) were distributed. Results were not stratified by kit received. | Baseline: 27% complaints of infections and irritation during last menstruation with usual item. Denominator not reported |
| Gade & Hytti 2017 [31] | Uganda, cohort, refugee camp, schoolgirls and women, 4 months, N = 102 | AFRIpad (4) | ~30% said that as a benefit of RMP, no irritation was felt using RMP; denominator not clear and it is not clear if this means 70% had irritation. | No information |
| Hooper 2020 [41] | India, cohort, schoolgirls (boarding), 12–24 m, N = 42 | RMP (brand NR) vs. menstrual cup | 16.7% pain and discomfort with RMP. Denominator not reported | 60.0% pain and discomfort with menstrual cup. Denominator not reported |
| Kuncio 2018 [37] (UNHCR) | Uganda, cohort, schoolgirls, refugee camps, 3 months, N = 273 | Afripad (4) at endline vs. usual item at baseline | Endline: 23.8% (40/267) had experienced itching or burning during last period when wearing AFRIpads (3-month follow-up). One girl stopped using AFRIpad because of this. | Baseline:72.8% (197/270) had experienced itching or burning when wearing disposable pads. No p-value reported. |
| Murthy 2015 [39] | India, cohort, rural women, 12–13 cycles, N = 45 | Uger | 40% (18/45) reported no discomfort such as itching and burning with Uger. It is not clear if this means 60% had irritation | No baseline information |
| Lenia 2019 [53] | Uganda, survey, women in refugee camp, NA, N = 422 | RMP (NR) vs disposable pads vs cloth | Adequate MHM practices† <br>• 50% (124/248) of RMP users (p = 0.001 compared to disposable pad users, p<0.001 compared to cloth users) | Adequate MHM practices† <br>• 65% (145/223) of disposable pad users <br>• 78% (54/69) of cloth users |
| Hennegan 2016 [25] | Uganda, endsurvey trial, schoolgirls, 12–24 months, N = 538 | AFRIpads (6) | Adequate MHM practices among RMP-users† <br>• 11.1% (8/72) of RMP users (p = 0.727 compared to usual practice) <br>Adequate MHM practices, relaxed criteria‡ <br>• 19.4% (14/72) of RMP users (p = 0.654 compared to usual practice) <br>• Wears usually damp pad: 6.9% (5/72, p = 0.007) | Adequate MHM practices among usual item† <br>• 8.5% (11/129) of usual practice <br>Adequate MHM practices, relaxed criteria‡ <br>• 23.3% (30/129) of usual practice <br>• Wears usually damp pad: 24.1% (20/83) |

MHM: Menstrual hygiene management. NR, not reported. RMP: Reusable menstrual pad.

*Uganda: Rhino refugee camp received kit A with 16 disposable pads; Mungula camp received kit B with reusable pads (3 winged pads and 5 straight pads). Madagascar received kits A & B to all communes. Somaliland received kit C with 10 disposable pads and 1 pack of reusable pads, quantity not specified. All kits contained underwear (2), use, care and disposal instruction for item, polyethylene storage bag, plastic bucket with lid, bar of personal bathing soap. Kits with disposable pads also contained biodegradable plastic bags. Kits with reusable pads also contained plastic coated rope and pegs and laundry soap.

†Adequate MHM practices: Clean menstrual management materials, change of materials at recommended intervals with privacy (3 times or more), use of soap and water for bathing and washing materials, and adequate disposal of material, drying of item outside.

sizes) of pads to start with, ranging from 6–27 RMPs depending on the duration and intensity of menstruation, and commonly suggest storing the soiled RMPs and washing them after their menstruation is finished using a washing machine (S1 File). Among 91 Ugandan schoolgirls

who experienced the benefits of a kit containing five RMPs in 2014, 52.3% reported they would not be able to afford it (lifespan of one year) if it had costed approximately 6.0 US$ [24]. Yilmaz et al (2019) examined whether Nepalese schoolgirls' willingness to pay for an RMP was affected by feeling the RMP material [65]; they were informed about RMPs either by reading a paragraph only or by being able to touch and feel the RMP. Girls in the latter group were willing to pay significantly more (15.8–24.6%) than girls who only read the paragraph [65].

## Costs and waste compared to other products

We stratified costs and waste over 2.5 and 5 years separately for LMIC and HIC (S1 File). Compared to tampons, single-use pads, and menstrual cups, cost-savings depended on the number of RMPs needed per menstruation and the lifespan of the RMPs, e.g., the longer the lifespan of the RMP and the fewer needed per cycle compared to single-use pads or tampons, the faster and higher the savings. If a woman in a LMIC used 8 RMPs with a life span of 2.5 years instead of 15 single-use pads per menstruation, she would spend 16.4 US$ and save approximately 135 US$ and avoid the waste of 488 single-use pads in 2.5 years; over 5 years this would amount to 278 US$ saved and avoid the waste of close to 1000 single-use pads (S1 File). With our cost estimate of single-use pads, over 5 years any number of RMPs examined (4–25) would be cheaper (170–417 US$) than 9–25 single-use pads per period with waste-savings of ~600–1600 single-use pads. If a woman in a HIC would use 8 RMPs with a life span of 5 years instead of 20 single-use tampons per menstruation, she would spend 81 US$ and save ~ 62 US$ and avoid the waste of 650 single-use tampons over 2.5 year; over 5 years it would save her approximately US$ 205 and avoid waste of 1300 single-use tampons. These calculations did not take additional costs for cleaning into account (e.g., water, soap, electricity of washing machine or time lost when washing) and did not look at production costs. Note that the mean price of a menstrual cup was estimated at 24 US$, and over a year would be cheaper than any number of single-use pads, or tampons, and cheaper than 8 RMPs over 5 year in LMIC and any number of RMPs in HIC [7]. In several studies, menstruating persons acknowledged the benefits of RMPs and the saving of money in the longer term; however, the initial costs were considered too steep to be a viable option [46,63].

## Discussion

In this review we aimed to summarize knowledge on RMPs in relation to experiences with use, menstrual blood leakage, and safety. All information on use was obtained from LMIC countries; RMPs were not mainstream, with low use if not delivered through a programme. Consumer satisfaction with RMPs appeared to be context-dependent, with higher approval in most impoverished settings, and lower approval in populations that have access to single-use menstrual products when sufficient resources are present [12]. Results from the studies implied a diversity of quality of RMPs (and single-use pads), impacting on uptake, acceptability, and safety. With regards to leaking, RMPs appeared to be more effective than cloths but RMPs did not consistently result in more mobility compared to the usual item (cloths or single-use pads). Difficulties with changing, washing, and drying of RMPs were reported to be recurring concerns: carrying used RMPs and washing off the menstrual blood can be perceived as unpleasant while washing takes time, water, soap, equipment and requires privacy. The lack of these necessities is likely to be of greater significance in LMICs than in HICs where access to washing machines is common.

Similar to menstrual cups [7], RMPs are not routinely included as choices in education materials for girls reaching menarche. RMP producers are present in both LMIC and HIC, but in HIC costs were higher with a reported longer average lifespan of the product. While not as

cost- and waste-saving as menstrual cups in the long-term, savings in costs and waste of RMPs are still considerable e.g., we estimated over 5 years one person would save ~278 US$ and reduce waste of ~ 1000 single-use pads in LMIC (comparing 15 single-use pads and 8 RMPs per period), or ~205 US$ and waste of ~ 1300 single-use tampons in HIC (comparing 20 single-use tampons and 8 RMPs per period). However, the higher upfront costs for RMPs could be a barrier for persons needing to purchase their own materials.

The most significant drawback of RMPs is that their successful use is largely dependent on the user's access and ability to wash, dry and maintain hygienic practices [66]. Washing is a barrier; some African participants noted seasonal problems, such as long drying times required in the rainy season, and lack of water in the dry season [31]. Others noted that reusable pads were more pleasant to wear, but single-use pads were more convenient to use [55]. Reusable menstrual materials have grown in popularity for distribution in emergencies, as these are perceived to be more sustainable and cost-effective; twenty percent of included studies were conducted in refugee camps or among vulnerable women [67]. However, a minimum number of RMPs and good sanitary conditions are required, to enable good menstrual practice, and avoid use of damp materials which may predispose to chaffing and sores. Despite the less optimal conditions in refugee camps, studies reported that women can successfully use RMPs, but some voiced a preference towards single-use products [33]. In HICs, environmental consciousness and comfort are drivers for the use of RMPs [44,64].

Included studies reported no obvious safety problems, except skin irritation associated with extended use of the same RMP, or with inadequate cleaning or drying of the RMP. No study reported on safety issues when comparing cloth and RMP use or had used objective methods of safety assessments instead of self-reporting by participants. Indeed, some participants reported using RMPs to avoid adverse effects of single-use pads [64], although others reported similar complaints to those arising from wearing cloths (chaffing, irritation, burning), albeit generally to a lesser extent. It is difficult to extrapolate how many women currently are using RMPs; however, the number of women who have received them through programmes is considerable and would make it likely that severe adverse events, if they were common, would be detected. *AfriPad* for example, reported that it produced as many as 30,000 RMPs per month in 2015, and in 2019 UNHCR reported it planned to distribute about 150,000 menstrual hygiene kits with RMPs [68]. In their annual report, *Days for Girls* reported it had distributed 362,500 menstrual kits in 2019 [69]. It is important that along with widespread distribution, efforts are made to objectively monitor any adverse events to ensure safety is clearly captured.

New RMPs are still being developed which is encouraging given that the ideal product for menstruation is determined by individuals' needs and their environmental setting; the recent expansion of new materials and methods to deal with menstrual blood are evident [70,71] (S1 File). New types of RMPs using silicone or polyester can contain menstrual blood within larger spaces in the material; the blood is washed out after use, and the pad can be dried with a towel, allowing the pad to be immediately available for reuse (S1 File). The wide variety of RMPs offered in HICs suggests there is sufficient demand to encourage manufacturers to continue to improve and diversify their products (S1 File). As part of this development, several countries are in the process of standardizing the requirements to manufacture RMPs [72]. Although this may improve the overall quality of RMPs, it can also limit e.g., environmental sustainability. In Uganda, manufacturers are obliged to add a protective barrier to the RMP, which usually takes the form of a plastic or polyester layer (PUL: polyurethane laminated fabric) to the RMP [73]. Some women may prefer to avoid these non-degradable protective barriers and choose to change RMPs more frequently or use a pad with compost-friendly materials. Locally-made RMPs can contribute to the local economy, as described in several papers and reports [28,74], and may lead to a better distribution of wealth than single-use pads produced by a few large

corporations. Although not as cost-and waste saving as the menstrual cup, the savings in waste when using RMPs can be considerable, e.g., 1000 single-use pads or tampons in 5 years (S6 Fig in S1 File). Currently, visibility and availability of RMPs is limited and mainly through online sources. It would be useful if this could be expanded to other avenues such as supermarkets and department stores in order to improve access and use.

## Limitations

The quality of the studies was generally low, with insufficient details available to meta-analyse outcomes. There was insufficient numeric data and no systematic data on safety. The number of studies from HIC was limited. The results of the web search of RMPs can only be considered as a snapshot or sample of what was available in the English language in 2020 because of limitations in our search (we did not include facebook, linkedln or Instagram for example) and a high turnover or name changes of RMP brands. With the increasing attention to menstrual health, countries are collecting more data on menstruation in national surveys. It was disappointing to note that the type of information collected did not discriminate between single-use or reusable pads, such that national survey data could not be included in this review [75]. We did not include studies on homemade RMPs. These can be of varying quality and production depends on time, equipment and the producer's dexterity. However, homemade pads can suit some persons well, and positive experiences have been reported [76,77]. Designs and instructions are available on websites, (e.g. [78]). For the cost-estimations, we made a difference in RMP prices for LMIC and HIC; it is possible that costs for single-use pads in LMIC are cheaper than our estimates.

## Public health impact

Given the low coverage of RMPs in education material for menarche (39%), and their low physical presence in stores, it is clear that many women and girls and programmes will not be aware of RMPs. Additionally, RMPs have high upfront costs and need a minimum level of sanitation for maintenance. Disposal of menstrual waste is often neglected when considering menstrual needs, but improper disposal of menstrual waste can lead to environmental pollution and clogging of sanitation systems (pit latrines or sewage systems) [79]. At the national level, countries can consider subsidizing purchases for the items with higher upfront costs, such as RMPs, make them free, or as a minimum remove tax, keeping in mind that a combination of different options may work best for an individual (e.g., combining a menstrual cup for heavier menstruation and RMPs for light days).

## Conclusion

This systematic review suggests that RMPs can be an alternative, effective, safe, cheaper, and environmentally friendly option for menstrual product provision by programmes. Further studies are needed e.g., in HIC, and when using more objective measures on safety, and to examine facilitators for use of RMPs. Improving knowledge about, and access to, different menstrual products will enable all persons who menstruate to make informed choices, impacting their health and quality of life.

## Supporting information

**S1 PRISMA checklist.**
(DOCX)

**S1 File. Supplement.**
(PDF)

## Acknowledgments

We would like to thank Shivali Bagayatka for her help with finding more websites of reusable pads in India, and Grace Francoise Nibizi for informing us about Agateka. We are grateful to Julie Hennegan for providing additional information for one of the studies she was involved in.

## Author Contributions

**Conceptualization:** Anna Maria van Eijk, Penelope A. Phillips-Howard.

**Data curation:** Anna Maria van Eijk, Naduni Jayasinghe, Garazi Zulaika, Linda Mason.

**Formal analysis:** Anna Maria van Eijk, Naduni Jayasinghe, Garazi Zulaika, Linda Mason.

**Funding acquisition:** Penelope A. Phillips-Howard.

**Investigation:** Muthusamy Sivakami, Holger W. Unger.

**Methodology:** Anna Maria van Eijk, Naduni Jayasinghe, Garazi Zulaika, Linda Mason, Muthusamy Sivakami, Holger W. Unger.

**Project administration:** Anna Maria van Eijk.

**Supervision:** Penelope A. Phillips-Howard.

**Writing – original draft:** Anna Maria van Eijk, Naduni Jayasinghe, Garazi Zulaika, Linda Mason.

**Writing – review & editing:** Muthusamy Sivakami, Holger W. Unger, Penelope A. Phillips-Howard.

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
