## [Decision Letter · Decision Letter 0]

21 Jul 2021

PONE-D-21-11102

Exploring menstrual products: a systematic review and meta-analysis of reusable menstrual pads for public health internationally

PLOS ONE

Dear Dr. Anna Maria Van Eijk

Thank you for submitting your manuscript to PLOS ONE. After careful consideration, we feel that it has merit but does not fully meet PLOS ONE’s publication criteria as it currently stands. Therefore, we invite you to submit a revised version of the manuscript that addresses the points raised during the review process.

We look forward to receiving your revised manuscript.

Kind regards,

Balasubramani Ravindran, Ph.D

Academic Editor

PLOS ONE

2. Please include the tables within the main manuscript.

“Joint Global Health Trials Initiative (UK-Medical Research Council/Department for International Development/Wellcome Trust/Department of Health and Social Care, grant MR/N006046/1)**”**

“Joint Global Health Trials Initiative (UK-Medical Research Council/Department for International Development/Wellcome Trust/Department of Health and Social Care, grant MR/N006046/1)**”**

Additional Editor Comments (if provided):

Reviewers' comments:

Reviewer's Responses to Questions

**Comments to the Author**

1. Is the manuscript technically sound, and do the data support the conclusions?

Reviewer #1: Partly

Reviewer #2: Yes

2. Has the statistical analysis been performed appropriately and rigorously? 

Reviewer #1: No

Reviewer #2: Yes

3. Have the authors made all data underlying the findings in their manuscript fully available?

Reviewer #1: No

Reviewer #2: Yes

4. Is the manuscript presented in an intelligible fashion and written in standard English?

Reviewer #1: Yes

Reviewer #2: Yes

5. Review Comments to the Author

Reviewer #1: A highlighted research area needs to public health was reviewed by the authors. Need some experimental data for supporting this review paper. The authors collected more numbers of research papers and consisted as a informative review.

Reviewer #2: The manuscript entitled "Exploring menstrual products: a systematic review and meta-analysis of reusable

menstrual pads for public health internationally" by Anna Maria van Eijk et al., is very well-written and summarizes the current status of the reusable menstrual products (RMPs). Given the demand for the eco-friendly and biodegradable alternatives, menstrual pads and diapers compose a huge portion of dumped thrash, posing serious hazard to the environment. An awareness about the availability of reusable alternatives to the single-use menstrual pads is need of the hour. This review summarizes the use, and limitations of the RMPs, incorporating statistical analysis of the demographically diverse women and transgenders. Such reviews will be of great use in highlighting the improvements needed for the future RMP products and I recommend publication of this article as is

Thanks

Vinuselvi

6. PLOS authors have the option to publish the peer review history of their article (what does this mean?). If published, this will include your full peer review and any attached files.

Reviewer #1: No

Reviewer #2: No

---

## [Author Response · Author response to Decision Letter 0]

4 Aug 2021

PONE-D-21-11102

Exploring menstrual products: a systematic review and meta-analysis of reusable menstrual pads for public health internationally

PLOS ONE

Dear Dr Ravindran,

We would like to thank you and the reviewers for their comments. Please find below our responses in italic. 

We hope the revised manuscript now meets the standards for publication.

Sincerely, for the authors,

Anna Maria (Annemieke) van Eijk

Response: The manuscript has been reformatted following the guidelines of the journal. 

2. Please include the tables within the main manuscript.

Response: The tables have been included within the main manuscript.

“Joint Global Health Trials Initiative (UK-Medical Research Council/Department for International Development/Wellcome Trust/Department of Health and Social Care, grant MR/N006046/1)”

“Joint Global Health Trials Initiative (UK-Medical Research Council/Department for International Development/Wellcome Trust/Department of Health and Social Care, grant MR/N006046/1)”

Response: We removed the funding-section at the end of the text of the manuscript, which read as follows: 

“Funding

This study is funded by the Joint Global Health Trials Initiative (UK-Medical Research Council/Department for International Development/Wellcome Trust/Department of Health and Social Care, grant MR/N006046/1). The funders had no role in the design of the study, the collection, analysis, and interpretation of data, or in writing the manuscript. The corresponding author had full access to all data in the study and had final responsibility to submit for publication.”

Response: It would be nice if the section below could be added to that statement; however, we do not think this essential if that would be a problem.

“The funders had no role in the design of the study, the collection, analysis, and interpretation of data, or in writing the manuscript. The corresponding author had full access to all data in the study and had final responsibility to submit for publication.”

Response: We have reviewed the reference list for completeness, following the Plos One guidelines and using the Plos One Endnote template. We removed Wilson et al (2019), because this manuscript could not be retrieved in a published or accepted version. The text, figures and tables were updated accordingly.

Reviewers' comments: 

Review Comments to the Author

5) Reviewer #1: A highlighted research area needs to public health was reviewed by the authors. Need some experimental data for supporting this review paper. The authors collected more numbers of research papers and consisted as a informative review.

• Outstanding review work was done by the authors. A lot of assortment date related to exploring menstrual products.

• These review paper, authors done a systemic review related to reusable menstrual pads.

• The topic is very highlighted and useful for public health, the authors collected and reviewed quantitative and qualitative studies that reported on leakage, acceptability, or safety of RMPs.

Response: We are pleased that the reviewer was positive about the manuscript, and we thank the reviewer for the time spent on this. The reviewer request for experimental data made no sense in the context of this systematic review and meta-analysis, where data pertaining to the study aims and objectives were fully extracted, analysed, and interpreted. As such, we have subsequently not made any changes based on these comments. 

6) Reviewer #2: The manuscript entitled "Exploring menstrual products: a systematic review and meta-analysis of reusable menstrual pads for public health internationally" by Anna Maria van Eijk et al., is very well-written and summarizes the current status of the reusable menstrual products (RMPs). Given the demand for the eco-friendly and biodegradable alternatives, menstrual pads and diapers compose a huge portion of dumped thrash, posing serious hazard to the environment. An awareness about the availability of reusable alternatives to the single-use menstrual pads is need of the hour. This review summarizes the use, and limitations of the RMPs, incorporating statistical analysis of the demographically diverse women and transgenders. Such reviews will be of great use in highlighting the improvements needed for the future RMP products and I recommend publication of this article as is

Response: We thank the reviewer for the positive comments and have not made any changes based on these comments.

Response: The figures have been uploaded as requested. Note that Figure 1 (the flow chart) was adapted to reflect the exclusion of the Wilson 2019 reference.

---

## [Decision Letter · Decision Letter 1]

7 Sep 2021

Exploring Menstrual products: a Systematic Review and Meta-analysis of Reusable Menstrual Pads for Public Health Internationally

PONE-D-21-11102R1

Dear Dr. Anna Maria Van Eijk,

We’re pleased to inform you that your manuscript has been judged scientifically suitable for publication and will be formally accepted for publication once it meets all outstanding technical requirements.

Kind regards,

Balasubramani Ravindran, Ph.D

Academic Editor

PLOS ONE

Additional Editor Comments (optional):

Reviewers' comments:

Reviewer's Responses to Questions

**Comments to the Author**

1. If the authors have adequately addressed your comments raised in a previous round of review and you feel that this manuscript is now acceptable for publication, you may indicate that here to bypass the “Comments to the Author” section, enter your conflict of interest statement in the “Confidential to Editor” section, and submit your "Accept" recommendation.

Reviewer #1: All comments have been addressed

2. Is the manuscript technically sound, and do the data support the conclusions?

Reviewer #1: Partly

3. Has the statistical analysis been performed appropriately and rigorously? 

Reviewer #1: Yes

4. Have the authors made all data underlying the findings in their manuscript fully available?

Reviewer #1: Yes

5. Is the manuscript presented in an intelligible fashion and written in standard English?

Reviewer #1: Yes

6. Review Comments to the Author

Reviewer #1: Manuscript once again check with grammatical error and language improvement and will be accept for publication.

7. PLOS authors have the option to publish the peer review history of their article (what does this mean?). If published, this will include your full peer review and any attached files.

Reviewer #1: No

---

## [Editor Report · Acceptance letter]

16 Sep 2021

PONE-D-21-11102R1 

Exploring Menstrual Products: a Systematic Review and Meta-analysis of Reusable Menstrual Pads for Public Health Internationally 

Dear Dr. van Eijk:

I'm pleased to inform you that your manuscript has been deemed suitable for publication in PLOS ONE. Congratulations! Your manuscript is now with our production department. 

Kind regards, 

on behalf of

Dr. Balasubramani Ravindran 

Academic Editor

PLOS ONE